# Differentiable Mathematical Programming for Object-Centric Representation Learning

**Adeel Pervez**
QUVA Lab,
Informatics Institute
University of Amsterdam
a.a.pervez@uva.nl

**Phillip Lippe**
QUVA Lab,
Informatics Institute
University of Amsterdam
p.lippe@uva.nl

**Efstratios Gavves**
QUVA Lab,
Informatics Institute
University of Amsterdam
e.gavves@uva.nl

## Abstract

We propose topology-aware feature partitioning into $k$ disjoint partitions for given scene features as a method for object-centric representation learning. To this end, we propose to use minimum $s$-$t$ graph cuts as a partitioning method which is represented as a linear program. The method is topologically aware since it explicitly encodes neighborhood relationships in the image graph. To solve the graph cuts our solution relies on an efficient, scalable, and differentiable quadratic programming approximation. Optimizations specific to cut problems allow us to solve the quadratic programs and compute their gradients significantly more efficiently compared with the general quadratic programming approach. Our results show that our approach is scalable and outperforms existing methods on object discovery tasks with textured scenes and objects

## 1 Introduction

Object-centric representation learning aims to learn representations of individual objects in scenes given as static images or video. Object-centric representations can potentially generalize across a range of computer vision tasks by embracing the compositionality inherent in visual scenes arising from the interaction of mostly independent entites. (Burgess et al., 2019; Locatello et al., 2020; Elsayed et al., 2022). One way to formalize object-centric representation learning is to consider it as an input partitioning problem. Here we are given a set of spatial scene features, and we want to *partition* the given features into $k$ per-object features, or slots, for some given number of objects $k$. A useful requirement for a partitioning scheme is that it should be *topology-aware*. For example, the partitioning scheme should be aware that points close together in space are often related and may form part of the same object. A related problem is to *match* object representations in two closely related scenes, such as frames in video, to learn *object permanence* across space and time.

In this paper we focus on differentiable solutions for the partitioning and matching problems that are also efficient and scalable for object-centric learning. We formulate the topology-aware $k$-part partitioning problem as the problem of solving $k$ *minimum $s$-$t$ cuts* in the image graph (see Figure 1) and the problem of matching as a *bipartite matching* problem.

An interesting feature of the minimum $s$-$t$ cut and bipartite matching problems is that they can both be formulated as linear programs. We can include such programs as layers in a neural network by parameterizing the coefficients of the objective function of the linear program with neural networks. However, linear programs by themselves are not continuously differentiable with respect to the objective function coefficients (Wilder et al., 2019). A greater problem is that batch solution of linear programs using existing solvers is too inefficient for neural network models, especially when the programs have a large number of variables and constraints. We solve these problems by 1) approximating linear programs by regularized equality constrained quadratic programs, and 2) pre-computing the optimality condition (KKT matrix) factorizations so that optimality equations can be quickly solved during training.

The advantage of using equality constrained quadratic programs is that they can be solved simply from the optimality conditions. Combined with the appropriate precomputed factorizations for the task of object-centric learning, the optimality conditions can be solved very efficiently during training.

---

**Algorithm 1** Feature $k$-Part Partitioning

---

**Require:** Input features $x$ of dimension $C \times H \times W$ with $C$ channels, height $H$ and width $W$
1: Compute quadratic program parameters $y_i = f_y^i(x)$, for $i \in \{1, \ldots k\}$ where $f_q^i$ are CNNs.
2: Optionally transform spatial features $x_f = f_x(x)$, where $f_x$ is an MLP transform acting on the channel dimension. $x_f$ has dimension $D \times H \times W$.
3: Solve regularized quadratic programs for minimum $s$-$t$ cut and extract vertex variables $z_i = \text{qsolve}(y_i)$ for each $y_i$. Each $z_i$ has dimension $H \times W$.
4: Normalize $z_i$ across cuts $i = 1, ..., k$ for each pixel with a temperature-scaled softmax.
5: Multiply $z_i$ with $x_f$ along $H, W$ for each $i$ to obtain $K$ masked features maps $r_i$.
6: Return $r_i$ as the $k$-partition.

---

A second advantage of using quadratic programming approximations is that quadratic programs can be differentiated relative to program parameters using the implicit function theorem as shown in prior literature (Barratt, 2018; Amos and Kolter, 2017).

To learn the objective coefficients of the cut problem by a neural network, the linear program needs to be solved differentiably, like a hidden layer. For this, we can relax the linear program to a quadratic program and employ techniques from differentiable mathematical programming (Wilder et al., 2019; Barratt, 2018) to obtain gradients. This amounts to solving the KKT optimality conditions which then result in the gradients relative to the parameters of the quadratic program (Amos and Kolter, 2017; Barratt, 2018). However, with a naive relaxation, the required computations for both the forward and backward pass are still too expensive for use in object-centric representation learning applications.

Given that the techniques generally employed for differentiably solving quadratic programs are limited to smaller program sizes (Amos and Kolter, 2017), we introduce optimizations in the gradient computation specific to the problem of solving graph cuts for image data. For instance, we note that the underlying $s$-$t$ flow graph remains unchanged across equally-sized images, allowing us to pre-compute large matrix factorization. Furthermore, we replace the forward pass by a *regularized equality constrained quadratic program* constructed from the linear programming formulation of the minimum $s$-$t$ cut problem. When combined with these task specific optimizations, equality constrained quadratic programs can be solved significantly more efficiently than general quadratic programs with mixed equality and inequality constraints (Wright and Nocedal, 1999). The regularization of slack variables ensures that the output of the new quadratic program can still be interpreted as an $s$-$t$ cut solution. The use of sparse matrix computations in the forward and backward passes ensures that time and memory usage is significantly reduced.

To summarize, we make the following contributions in this paper.

1. We formulate object-centric representation learning in terms of partitioning and matching.
2. We propose $s$-$t$ cuts in graphs for topology-aware partitioning with neural networks.
3. We propose to use regularized equality constrained quadratic programs as a differentiable, general, efficient and scalable scheme for solving partitioning and matching problems with neural networks.

## 2 MINIMUM $s$-$t$ CUTS FOR TOPOLOGY-AWARE PARTITIONING

We first describe the general formulations of the graph partitioning problem specialized for images and then describe limitations when considering graph partitioning in image settings. With these limitations in mind, we describe the proposed *neural $s$-$t$ cut* and *matching* algorithms, which allow for efficient and scalable solving of graph partitioning and matching. Last, we describe how to learn end-to-end object-centric representations for static and moving objects with the proposed methods.

### 2.1 MINIMUM $s$-$t$ GRAPH CUTS

The problem of finding minimum $s$-$t$ cuts in graphs is a well-known combinatorial optimization problem closely related to the max-flow problem (Kleinberg and Tardos, 2005). We are given a directed graph $G = (V, E)$ with weights for edge $(u, v)$ denoted by $w_{u,v}$ and two special vertices

$s$ and $t$. The minimum $s$-$t$ cut problem is to find a partition of the vertex set $V$ into subsets $V_1$ and $V_2$, $V_1 \cap V_2 = \emptyset$, such that $s \in V_1$, $t \in V_2$ and the sum of the weights of the edges going across the partition from $V_1$ to $V_2$ is minimized. In classical computer vision, the minimum cut problem has been used extensively for image segmentation (Boykov and Jolly, 2001; Shi and Malik, 2000). The problem of image segmentation into foreground and background can be reduced to minimum $s$-$t$ cut by representing the image as a weighted grid graph with the vertices representing pixels (Figure 1) where the weights on the edges represent the similarity of neighbouring pixels. Next, the vertices $s$ and $t$ are introduced and for each pixel vertex $v$ we include edges $(s, v)$ and $(v, t)$. The weights on edges $w_{s,v}$, $w_{t,v}$ represent the relative background and foreground weight respectively for the pixel vertex $v$. For example, $w_{s,v} > w_{t,v}$ may indicate that $v$ is more likely to be in the foreground rather than the background. At the same time, the neighbouring edge weights $w_{u,v}$ indicate that similar pixels should go in the same partition. Solving the min-cut problem on this graph leads to a segmentation of the image into foreground and background. Note that this procedure encodes the underlying image topology in the graph structure.

**Solving Cut Problems.** Given a directed graph $G = (V, E)$ with edge weights $w_{u,v}$ and two special vertices $s$ and $t$, we can formulate the min-cut problem as a linear program (Dantzig and Fulkerson, 1955). To do so, we introduce a variable $p_u$ for each vertex $u$ in the linear program. Similarly, for each edge $(u, v)$, we introduce a variable denoted $d_{u,v}$. The edge weights $w_{u,v}$ act as objective function parameters in the linear program.

$$
\begin{aligned}
\text{minimize} \quad & \sum w_{uv} d_{uv} \\
\text{subject to} \quad & d_{uv} \geq p_u - p_v && (u, v) \in E, \\
& p_s - p_t \geq 1, \\
& d_{u,v} \geq 0 && (u, v) \in E, \\
& p_v \geq 0 && u \in V
\end{aligned}
\tag{1}
$$

The program can then be fed to a linear programming solver for the solution. Although there are other more efficient methods of solving min-cut problems, we resort to the linear programming formulation because of its flexibility and that it can be approximated with differentiable proxies.

## 2.2 TOPOLOGY-AWARE PARTITIONING WITH GRAPH CUTS

We propose finding minimum $s$-$t$ cuts in the image graph with weights parameterized by neural networks, $w_{u,v} = f(x)$ with image $x$, for partitioning image feature maps, $x' = g(x)$, into $k$ disjoint partitions. Solving the standard minimum $s$-$t$ cut problem can divide the vertex set of an input graph into two partitions. We generalize the minimum $s$-$t$ cut problem to partition a given graph into any fixed $k$ number of partitions by solving $k$ parallel min-cut problems and subsequently normalizing the vertex (or edge) variables to sum to 1 across the $k$ copies of the graph. We note that 2-partition can also be formulated as a subset selection problem (Xie and Ermon, 2019). The advantage of using minimum $s$-$t$ cuts for partitioning is that the method then depends on the underlying topology and neighbourhood relations of image data represented in the image graph.

Figure 1: (Top) Image graph for 2-partition with an $s$-$t$ cut and 3 parallel graphs for 3-partition (Below).

**Partitioning Generalization.** We generalize the minimum $s$-$t$ cut method for 2-partition to $k$ partitions by solving $k$ parallel 2-partition problems and normalizing the vertex variables $p_u^i$, $i$ denoting partition index, from the optimum solution of equation 1 with a softmax over $i$. We also experimented with solving the $k$-part partitioning problem with a single graph cut formulation. However, we found such a formulation to have high memory usage with increasing $k$ and limited ourselves to the parallel cut formulation which worked well for our experiments.

**Obstacles.** There are some obstacles in the way of including linear programs as hidden layers in neural networks in a way that scales to solving large computer vision problems. The first of these is that linear programs are not continuously differentiable with respect to their parameters. One solution to this is to use the quadratic programming relaxation suggested by Wilder et al. (2019). However, the gradient computation for general large quadratic programs requires performing large matrix factorization which is infeasible in terms of computing time and memory when working with image-scale data.

A second obstacle is that established quadratic programming solvers such as OSQP (Stellato et al., 2020) and Gurobi (Gurobi Optimization, LLC, 2022) run on CPU and cannot solve programs in large batches essential for fast neural network optimization. There have been attempts to solve general quadratic programs with GPU acceleration (Amos and Kolter, 2017). However, in practice, these attempts do not scale well to quadratic program solving in batch settings with the tens of thousands of variables and constraints that arise when working with image data.

## 3 DIFFERENTIABLE MATHEMATICAL PROGRAMMING FOR NEURAL $s$-$t$ CUT

We describe our proposed method and optimizations to differentiably solve approximations of the $s$-$t$ cut problems as quadratic programs. Our method depends on the fact that 1) quadratic program solutions, unlike linear programs, can be differentiated relative to the objective function parameters (Amos and Kolter, 2017; Wilder et al., 2019) and 2) that quadratic programs with equality constraints only can be solved significantly more quickly than general quadratic programs (Wright and Nocedal, 1999). Combining the method with specific optimizations allows to solve large batches of quadratic programs on GPU allowing us to scale to image data.

### 3.1 REGULARIZED EQUALITY CONSTRAINED QUADRATIC PROGRAMS

We use the fact that quadratic programs with equality constraints only can be solved much more quickly and easily than programs with mixed equality and inequality constraints. Given an equality constrained quadratic programming, we can directly solve it by computing the KKT matrix factorization and solving the resulting triangular system (Wright and Nocedal, 1999).

In general, linear or quadratic programs (such as those for $s$-$t$ cut) have non-negativity constraints for variables which cannot be represented in equality constrained quadratic programs. Instead, we use regularization terms in the objective function which ensure that variables remain within a reasonable range. After solving the regularized equality constrained program, the solutions can be transformed, for example by a sigmoid or softmax, to bring them within the required range.

We approximate the minimum $s$-$t$ cut linear program (equation 1) by the following quadratic program with equality constraints only.

$$
\begin{aligned}
\text{minimize} \quad & \sum w_{uv}d_{uv} + \gamma \sum_{u,v}(d_{uv}^2 + r_{uv}^2) + \gamma \sum_v p_v^2 \\
\text{subject to} \quad & d_{uv} - r_{uv} = p_u - p_v \qquad (u,v) \in E, \\
& p_s - p_t - r_{st} = 1,
\end{aligned}
\tag{2}
$$

where the variables $r_{uv}$ are slack variables and $\gamma$ is a regularization coefficient. We achieve this by first adding one slack variable $r_{uv}$ per inequality constraint to convert them into equality constraints plus any non-negativity constraints. Next, we add a diagonal quadratic regularization term for all variables, including slack variables in the objective function. Finally, we remove any non-negativity constraints to obtain a quadratic program with equality constraints only.

**Parameterization.** Given input features $x$, we can include regularized equality constrained programs for $s$-$t$ cut in neural networks by parameterizing the edge weights $w_{u,v}$ by a ConvNet $f$ as $w = f(x)$. Assuming the image graph in Figure 1 has height $H$ and width $W$, we can parameterize the weights for the partition by using a ConvNet $f$ with output dimensions $(6k, H, W)$ giving the weights for 6 edges (4 neighbor edges and $s$ and $t$ edges) per pixel per partition. Our experiments show that the output of the regularized equality-constrained quadratic program for minimum $s$-$t$ cut can easily be interpreted as a cut after a softmax and works well for image-scale data.

**Forward and backward pass computations.** We now describe the computations required for solving regularized equality constrained quadratic programs and their gradients relative to the objective function parameters. A general equality constrained quadratic program has the following form.

$$
\begin{aligned}
\text{minimize} \quad & \tfrac{1}{2}z^t Gz + z^t c \\
\text{subject to} \quad & Az = b,
\end{aligned}
\tag{3}
$$

where $A \in \mathbb{R}^{l \times n}$ is the constraint matrix with $l$ constraints, $b \in \mathbb{R}^l$, $G \in \mathbb{R}^{n \times n}$ and $c \in \mathbb{R}^n$. For our case of quadratic program proxies for linear programs, $G$ is a multiple of identity, $G = \gamma I$,

and corresponds to the regularization term. For our particular applications, $c$ is the only learnable parameter and $A$ and $b$ are fixed and encode the image graph constraints converted to equality constraints by use of slack variables. Any non-negativity constraints are removed.

$$
\begin{aligned}
\text{minimize} \quad & \sum c_{uv} d_{uv} \\
\text{subject to} \quad & p_u + d_{u,v} = 1, \\
& p_v + d_{u,v} = 1 \; \{u, v\} \in E, \\
& d_{u,v} \geq 0 \; \{u, v\} \in E
\end{aligned}
\tag{4}
$$

Figure 2: Linear program for matching

Figure 3: Schematic of partitioning module

The quadratic program can be solved directly by solving the following KKT optimality conditions for some dual vector $\lambda$ (Wright and Nocedal, 1999),

$$
\begin{bmatrix} G & A^t \\ A & 0 \end{bmatrix} \begin{bmatrix} -z \\ \lambda \end{bmatrix} = \begin{bmatrix} c \\ -b \end{bmatrix},
\tag{5}
$$

where the matrix on the left is the KKT matrix. We can write the inverse of the KKT matrix using Gaussian elimination (the Schur complement method, (Wright and Nocedal, 1999)), with $G = \gamma I$, as

$$
\begin{bmatrix} \gamma I & A^t \\ A & 0 \end{bmatrix}^{-1} = \begin{bmatrix} C & E \\ E^t & F \end{bmatrix},
\tag{6}
$$

where $C = \frac{1}{\gamma}(I - A^t(AA^t)^{-1}A)$, $E = A^t(AA^t)^{-1}$ and $F = -(AA^t)^{-1}$. We note that only $(AA^t)^{-1}$ appears in inverted form in equation 6 which does not change and can be pre-computed. We can pre-compute once using a Cholesky factorization to factor $AA^t$ as $AA^t = LL^t$ in terms of a triangular factor $L$. Then during training whenever we need to solve the KKT system by computing $Cz'$ or $Ez'$ (equation 6) for some $z'$, we can do this efficiently by using forward and backward substitution with the triangular factor $L$.

Once we have the KKT matrix factorization equation 6 we can reuse it for the gradient computation. The gradient relative to parameters $c$ is obtained by computing $\nabla_c l(c) = -C\nabla_z l(z)$, where $l(.)$ is a loss function and $z$ is a solution of the quadratic program (see Amos and Kolter (2017) for details). We can reuse the Cholesky factor $L$ for the gradient making the gradient computation very efficient.

**Sparse Matrices.** The final difficulty is that the constraint matrix $A$ can be very large for object-centric applications since the image graph scales quadratically with the image resolution. For 64x64 images we get $s$-$t$ cut quadratic programs with over 52,000 variables and over 23,000 constraints (resulting in a constraint matrix of size over 4GB) which have to solved in batch for each training example. Since the constraints matrices are highly sparse (about 6% non-zero elements for 64x64 images) we use sparse matrices and a sparse Cholesky factorization (Chen et al., 2008). During training we use batched sparse triangular solves for the backward and forward substitution to solve a batch of quadratic programs with different right hand sides in equation 5. Assuming a graph size of $N$, solving the quadratic program directly (with dense Cholesky factorization) has complexity about $O(N^3)$ and is even greater for general quadratic programs. With our optimizations this reduces to $O(Nn_z)$, where $n_z$ is the number of nonzero elements, which is the time required for a sparse triangular solve.

### 3.2 EXTENSION TO MATCHING

We can extend the regularized quadratic programming framework to the problem of matching. Given two sets of $k$-part partitions $P_1$ and $P_2$ we want to be able to match partitions in $P_1$ and $P_2$. We can treat this as a bipartite matching problem which can be specified using linear program 4. We denote nodes variables as $p_u$ for node $u$, edge variables for edge $\{u, v\}$ as $d_{u,v}$ and the edge cost as $c_{u,v}$.

Unlike the $s$-$t$ cut problems, the matching problems that we use in this paper are quite small with the bipartite graph containing no more than $2 \times 12 = 24$ nodes. For such small problems one could conceivably employ (after suitable relaxation (Wilder et al., 2019)) one of the CPU-based solvers or the parallel batched solution proposed by Amos and Kolter (2017). However, in this paper, we use the same scheme of approximating the regularized equality constrained quadratic programs that we

use for graph cut problems for the matching problem and leave the more exact solutions to further work. See Appendix C for details of the formulation of regularized quadratic programs for matching.

## 4  APPLICATION TO OBJECT-CENTRIC REPRESENTATION LEARNING

We now discuss how to use the specific $k$-part partitioning formulation described above for object-centric learning. Given scene features, $x$, of dimension $C \times H \times W$ as input, we transform the features into $k$ vectors or *slots* where each slot corresponds to and represents a specific object in the scene. The objective is that, by means of graph cuts that preserve topological properties of the pixel space, the neural network can learn to spatially group objects together in a coherent fashion, and not assign far-off regions to the same slot arbitrarily.

We use a similar encoder-decoder structure as Slot Attention (Locatello et al., 2020). Crucially, however, we replace the slot attention module by our min-cut layer. Specifically, based on scene features that are extracted by an CNN backbone, we compute $k$ minimum $s$-$t$ cut quadratic program parameters using a small CNN. We solve the $k$ quadratic programs and softmax-normalize the vertex variables. This gives us (soft) spatial masks $z_i$ normalized across $i$. We multiply the

---
**Algorithm 2** Slots Computation
---
**Require:** Input features $x$ of dimension $C \times H \times W$ with $C$ channels, height $H$ and width $W$
1:  Add position encoding to features $x$.
2:  Partition $x$ using Algorithm 1 to obtain $k$ partitions $r_i$ of dimensions $D \times H \times W$.
3:  Average $r_i$ across $H, W$.
4:  Transform each $r_i$ to get $s_i = f(r_i)$ with MLP $f$.
5:  Return slots $S = \{s_1, ..., s_k\}$.

---

masks $z_i$ with the (optionally transformed) scene features $x$ broadcasting across the channel dimension. This gives us masked features $r_i$ of dimensions $D \times H \times W$ that form a partition. The procedure is described in Algorithm 2. To obtain slot object representations, we add position encoding to the scene features $x$ before partitioning. After partitioning, we average each partition $r_i$ of dimension $D \times H \times W$ across the spatial dimensions to obtain $k$ vectors of dimension $D$ which are then transformed by MLPs to obtain the slots. Finally, each slot is separately decoded with a separate output channel to combine the individual slot images. This combined image is then trained to reconstruct the original scene $x$. We make no changes to the reconstruction loss function in (Locatello et al., 2020)

**Background slot.** The basic object-centric framework described in Algorithms 1 and 2 can easily be extended to work with a specialized background slot. We do this by computing one set of masks for a 2-partition and one for $k-1$-part partition, denoted `mask_fg`, `mask_bg`, `mask_object`, and multiplying the foreground and object masks as `mask_fg*mask_object`. In experiments, we find that computing slots in this way specializes a slot for the background.

**Matching and motion.** We extend the model for learning representations of moving objects in video by combining the model with the matching procedure described in Algorithm 3. We work with pairs of frames. For each frame, we compute $k$ per-frame slot representations as before. Next using the matching procedure we match the two sets of slots to get new slots $r_i$ as in Algorithm 3, and transform the new slots by MLP transformations. The new slot is used in the decoder to reconstruct the input *pair* of frames by optimizing the sum of the reconstruction error for the two frames.

---
**Algorithm 3** Matching slots for scene pairs
---
**Require:** Input features $x_1, x_2$ of dimension $C \times H \times W$ with $C$ channels, height $H$ and width $W$
1:  Compute slots $S, T$ for $x_1, x_2$ respectively using Algorithm 2.
2:  Compute a cost for each pair of slots $s_i \in S$ and $t_i \in T$ by computing inner products as $c_{i,j} = -\langle s_i, t_j \rangle$.
3:  Solve matching using method in 3.2 to obtain matching matrix $M_{i,j}$.
4:  Apply softmax with temperature to $M_{i,j}$ along dimension $j$.
5:  Compute paired slot $r_i = M_{i,:}T$
6:  Return $k$ slots pairs $(s_i, r_i)$.

---

## 5  RELATED WORK

**Partitioning.** The problem of partitioning input data frequently appears in computing tasks. One commonly used approach casts the problem as one of selecting subsets. An input $x$ (a vector or sequence) is treated as a set where each element of the set is assigned a weight. Next, one can select $k$ elements with the largest weight using one of the available relaxation of the Top-$k$ operator (Plötz and Roth, 2018; Grover et al., 2018). If a stochastic sample is desired then the weights might be treated as unnormalized probabilities and the Top-$k$ relaxation combined with a ranking distribution such as

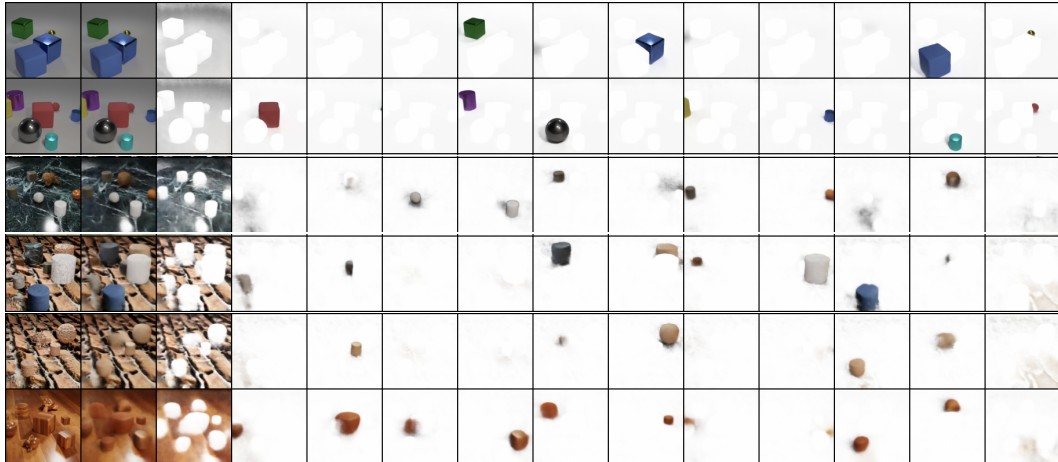

Figure 4: Unsupervised object discovery on Clevr, ClevrTex and CAMO. Columns (left to right) show original image, overall reconstruction and per-slot reconstruction for each of 12 slots.

Gumbel Softmax to obtain differentiable samples without replacement (Xie and Ermon, 2019). Or one might sample elements independently and employ a regularization method such $L_0$ regularization (Louizos et al., 2018; De Cao et al., 2020) to control the number selected elements. For problems involving partitions of images, the selection of a pixel makes it likely that neighboring pixels will also be chosen. Current methods often treat the problem as one of independent choice and rely on learning.

**Graph Cuts in Vision.** Both directed and undirected version of graph cuts have been considered previously for computer vision applications including image segmentation (Boykov and Jolly, 2001; Wu and Leahy, 1993; Boykov and Funka-Lea, 2006), image restoration (Greig et al., 1989), multi-view reconstruction (Kolmogorov and Zabih, 2001). For image segmentation with global cuts, normalized cut Shi and Malik (2000) has been a popular method to prevent the method from cutting small isolated sets. The work Boykov and Jolly (2001) uses $s$-$t$ cuts which is the approach taken in this work.

**Object-Centric Representation Learning.** Currently, most works on object-centric representation learning deploy a slot-based approach (Locatello et al., 2020). In this, the input is represented by a set of feature vectors, also called *slots*, where each slot encodes a different object. For instance, Slot Attention (Locatello et al., 2020) uses an iterative attention mechanism to encode an input image into slots. The model is trained via reconstruction loss by decoding each slot is separately, and combining the image via an alpha channel afterwards. Recent works extend Slot Attention to video data (Kipf et al., 2022; Elsayed et al., 2022; Kabra et al., 2021). Another group of works on slot-based object-centric representation learning explicitly split the input image into multiple parts by predicting one mask per object (Burgess et al., 2019; Engelcke et al., 2019; Greff et al., 2019; Engelcke et al., 2021). However, all previously mentioned works do not exploit the topology of the input image to partition it into objects. For this, Lin et al. (2020); Jiang and Ahn (2020); Crawford and Pineau (2019) predict bounding-boxes to locate objects that are reconstructed on top of a background slot. Alternatively to slots, Löwe et al. (2022); Reichert and Serre (2014) use complex-valued activations to partition the feature space via phase (mis-)alignments, and Smirnov et al. (2021); Monnier et al. (2021) learn recurring sprites as object prototypes.

## 6 EXPERIMENTS

### 6.1 OBJECT DISCOVERY

**Model.** We use the architecture in Locatello et al. (2020) and replace the slot attention mechanism by the $k$-partition module with background partitioning as described in Section 4. The architecture is an encoder-decoder architecture with a standard CNN (or ResNet) as the encoder and a spatial broadcast decoder. We use a CNN with 64 features maps in the encoder per layer for Clevr and a ResNet with 4 blocks with 100 features maps for ClevrTex. The feature maps are downsampled in the first layer to 64x64. The decoder in both cases is the same which broadcasts each slot across an 8x8 grid and upsamples to the input dimensions. We use a 3-layer convNet in the $k$-partition module to output the

Table 1: Object discovery performance on Clevr, ClevrTex with evaluation on OOD and CAMO (Karazija et al., 2021). Boldface indicates best metric. Underlined scores indicate second best.

| Model | Clevr | | | ClevrTex Textured Object+BG | | | OOD (eval-only) Out-of-dist. shape/texture | | | CAMO (eval-only) Camouflaged objects | | |
|---|---|---|---|---|---|---|---|---|---|---|---|---|
| | mIoU | ARI-FG | MSE | mIoU | ARI-FG | MSE | mIoU | ARI-FG | MSE | mIoU | ARI-FG | MSE |
| **Glimpse-based Methods** | | | | | | | | | | | | |
| SPAIR (Crawford and Pineau, 2019) | **65.95** | 77.13 | 55 | 0.0 | 0.00 | 1101 | 0.0 | 0.00 | 1166 | 0.0 | 0.0 | 668 |
| SPACE (Lin et al., 2020) | 26.31 | 22.75 | 63 | 9.14 | 17.53 | 298 | 6.87 | 12.71 | 387 | 8.67 | 10.55 | 251 |
| GNM (Jiang and Ahn, 2020) | 59.92 | 65.05 | 43 | 42.25 | 53.37 | 383 | **40.84** | 48.43 | 626 | 17.56 | 15.73 | 353 |
| **Sprite-based Methods** | | | | | | | | | | | | |
| MN (Smirnov et al., 2021) | 56.81 | 72.12 | 75 | 10.46 | 38.31 | 335 | 12.13 | 37.29 | 409 | 8.79 | 31.52 | 265 |
| DTI (Monnier et al., 2021) | 48.74 | 89.54 | 77 | 33.79 | 79.90 | 438 | 32.55 | **73.67** | 590 | 27.54 | 72.90 | 377 |
| **Pixel-Space Methods** | | | | | | | | | | | | |
| Gen. V2 (Engelcke et al., 2021) | 9.48 | 57.90 | 158 | 7.93 | 31.19 | 315 | 8.74 | 29.04 | 539 | 7.49 | 29.60 | 278 |
| eMORL (Emami et al., 2021) | 21.98 | 93.25 | 26 | 30.17 | 45.00 | 347 | 25.03 | 43.13 | 546 | 19.53 | 42.34 | 315 |
| MONet (Burgess et al., 2019) | 30.66 | 54.47 | 58 | 19.78 | 36.66 | **146** | 19.30 | 32.97 | **231** | 10.52 | 12.44 | **112** |
| IODINE (Greff et al., 2019) | 45.14 | 93.81 | 44 | 29.16 | 59.52 | 340 | 26.28 | 53.20 | 504 | 17.52 | 36.31 | 315 |
| Slot Attention (Locatello et al., 2020) | 36.61 | **95.89** | 23 | 22.58 | 62.40 | 254 | 20.98 | 58.45 | 487 | 19.83 | 57.54 | 215 |
| Ours | 49.04 | 95.40 | **14** | **42.4** | **80.2** | 207 | 38.0 | 72.6 | 577 | **38.9** | **75.5** | 189 |

parameters of the quadratic program. The partitioning module works with 64x64-size feature maps. The image features are transformed by MLPs before applying the partitioning and again after the partition. We use a fixed softmax temperature of 0.1 for the combined slots and a temperature of 0.5 for the foreground slots. We set the regularization parameter $\gamma = 0.5$.

**Datasets.** We experiment with the Clevr (Johnson et al., 2017) and ClevrTex datasets (Karazija et al., 2021). For Clevr we use the version with segmentation masks [1]. ClevrTex is a more complex version of Clevr with textured objects and background. ClevrTex also provides two evaluation sets, OOD: an out-of-distribution dataset with textures and shapes not present in the training set and CAMO: a camouflaged version with the same texture used for the objects and background. It was shown in (Karazija et al., 2021) that current methods perform poorly with the presence of textures and performance is drastically worse on the CAMO evaluation set. We crop and resize the images to 80% of the smaller dimension and taking the center crop. We then resize the images to 128x128. This is the same setup as used by (Karazija et al., 2021)

**Metrics.** For comparison of segmentation performance, the adjusted rand index with foreground pixels (ARI-FG) only has been used in prior work. It has been suggested (Karazija et al., 2021) that mean intersection over union (mIoU) might be a better metric since it also considers the background. Thus in this paper, we report both the foreground ARI and mIoU for the models.

**Results.** The results are shown in Table 1. For Clevr, we find that the segmentation performance for our method exceeds all other pixel-space methods in terms of mIoU. In terms of foreground ARI, our method matches the best performing model on this metric, namely Slot Attention, being only slightly worse. Finally, our method constitutes the best reconstruction performance on CLEVR. Example reconstructions from the validation set can be seen in Fig 4. The figure shows that our method is able to achieve very sharp reconstructions on CLEVR.

On the full ClevrTex dataset, our method outperforms all baselines in both mIoU and ARI-FG, providing an improvement of 12% and 18% respectively compared to the best pixel-space baseline. Furthermore, the strong reconstruction quality in terms of the background pattern in Fig 4 result in a low reconstruction loss. This shows the benefit of topology-aware partitioning for objects with complex textures.

**Generalization.** To test the generalization ability of our model, we took the model trained on the ClevrTex dataset and evaluated it on the OOD and CAMO evaluation datasets bundled with ClevrTex. The results are shown in Table 1. While all models experience a decrease in segmentation performance, our method remains one of the best to generalize. Especially on the CAMO dataset, our method outperforms all baselines in both mIoU and ARI-FG, even increasing the gap the best pixel-space baseline in terms of mIoU to 19%. The reason for this is that the min-cut problem can easily take into account the sharp transitions in the image between the background and foreground objects by increasing the weight for the edges that connect the pixels at the transitions. Other pixel-space

---

[1] https://github.com/deepmind/multi_object_datasets

methods like Slot Attention often focus on the color and texture for partitioning the image (Karazija et al., 2021; Dittadi et al., 2022). This makes the CAMO dataset much more difficult since all objects are almost identical in these attributes to the background.

## 6.2 MATCHING VALIDATION

Next we validate the matching procedure by using the experiment for bipartite matching with learned weights used by Wilder et al. (2019) with the Cora citation dataset. The task is to predict the presence of an edge only using the node features. A neural network predicts the parameters of the matching linear program which is relaxed to a quadratic program in Wilder et al. (2019) for training. We replace the relaxed quadratic program with a regularized equality constrained quadratic program with $\gamma = 0.1$. The main difference is that we do not have non-negativity constraints for the variables and use regularization instead. For evaluation we evaluate using the true linear program with the learned parameters using an LP solver. For this experiment we achieve an average (maximization) objective function value of 7.01±1.15 which is an improvement over 6.15±0.38 reported by Wilder et al. (2019).

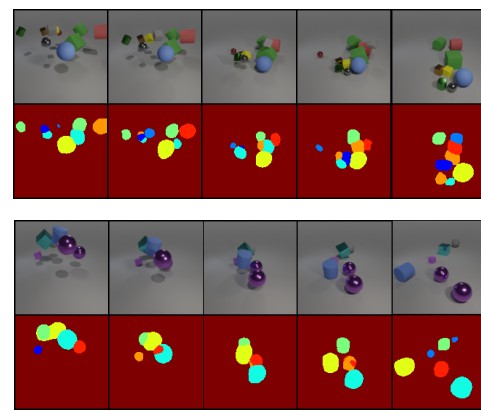

Figure 5: Capturing moving object on the MOVi-A dataset (without cropping). Top row shows original objects, second row shows object masks

**Qualitative Validation.** We also validate the matching method qualitatively on the MOVi-A dataset. We take two far separated frames and reconstruct the pair of frames using a slot matching as described in Algorithm 3. In this test we take frame index 5 and 20. An example is shown in Figure 6 in Appendix A where we show reconstructions for the two separated frames that have been successfully matched even though some objects are far from their original positions.

Table 2: Evaluation results on MOVi-A

| Model | ARI-FG (%) | ARI (%) |
|---|---|---|
| Slot Attention | 4.2 | 0.3 |
| Ours | **65.6** | **77.7** |

## 6.3 MOVING OBJECTS

We work with the MOVi-A (Greff et al., 2022) of moving object videos with 24 frames in each video. We crop the frames to 0.8 times a side, take the center and resize to 128x128 and use 8 slots per frame. We train the object discovery model on pairs of adjacent frames. Given pairs of scene features the slots module produces two sets of slots which are then matched together using the method described in Algorithm 3 into a single set of slots. The new set of slots is then used to reconstruct the two frames. Reconstructions from the background slot and 3 object slots (identified by position) are shown in Figure 5. The slot is able to track the object over multiple steps.

We compare against a slot attention baseline where we train slot attention on individual frames. Next, during evaluation, we match sets of slots for consecutive frames using the Hungarian algorithm, using inter slot inner product as the similarity score. The results are shown in Table 2. We see from the results that slot attention has no better than random clustering performance, whereas our cut-based method obtains a foreground ARI of 65.6 even though it was trained only with pairs of frames.

## 7 CONCLUSION

We propose a method for object-centric learning with partitioning and matching. We also present an efficient and scalable mathematical programming approach based on quadratic programming approximation for solving the partitioning and matching problems. In the experiments we showed that our approach can be used in both static and moving object settings. We also show that the method outperforms competing methods on object discovery tasks in textured settings [2].

---

[2]Code repository: `https://github.com/alpz/graph-ocl`

## 8 ACKNOWLEDGEMENTS

This work is financially supported by Qualcomm Technologies Inc., the University of Amsterdam and the allowance Top consortia for Knowledge and Innovation (TKIs) from the Netherlands Ministry of Economic Affairs and Climate Policy.

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

## A  MATCHING VALIDATION

We validate the matching method qualitatively on the MOVi-A dataset. We take two far separated frames and reconstruct the pair of frames using a slot matching as described in Algorithm 3. In this test we take frame index 5 and 20. An example is shown in figure 6 where we show reconstructions for the two separated frames that have been successfully matched even though some objects are far from their original positions.

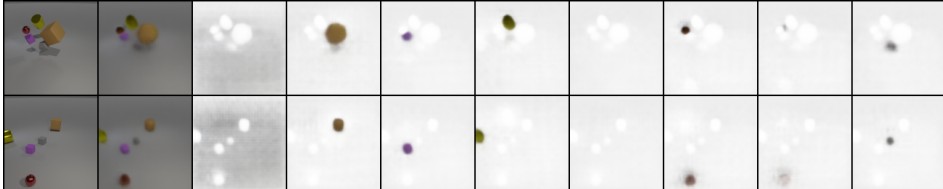

Figure 6: Matching far separated frames.

## B  SPARSE MATRIX COMPUTATIONS

The constraint matrix $A$ can be very large for object-centric applications since the image graph scales quadratically with the image resolution. For 64x64 images we get $s$-$t$ cut quadratic programs with over 52,000 variables and over 23,000 constraints (resulting in a constraint matrix of size over 4GB) which have to solved in batch for each training example. Since the constraints matrices are highly sparse (about 6% non-zero elements for 64x64 images) we use sparse matrices and a sparse Cholesky factorization (Chen et al., 2008). During training we use batched sparse triangular solves for the backward and forward substitution to solve a batch of quadratic programs with different right hand sides in equation 5. Since current neural network frameworks do not support sparse triangular solves we use CuPy (Okuta et al., 2017) which provides an interface to cuSPARSE (Nvidia, 2014). These optimizations result in fast computation of batches of quadratic programs and allow us to scale to image scale data. Assuming a graph size of $N$, solving the quadratic program directly (with dense Cholesky factorization) has complexity about $O(N^3)$ and is even greater for general quadratic programs. With our optimizations this reduces to $O(Nn_z)$, where $n_z$ is the number of nonzero elements which is the time required for a sparse triangular solve.

## C  REGULARIZED EQUALITY CONSTRAINED QUADRATIC PROGRAM FOR MATCHING

We approximate problem 4 by adding an extra slack variable to the constraints, remove the non-negativity constraint and add regularization terms to the objective. After solving the new quadratic program, we normalize the edge variables by using a softmax over the second vertex. Note that this does not give a perfect matching of $k$-partitions in the bipartite graph since multiple nodes in the first partition can be associated with a single node in the second. However, the regularization keeps the number of such multiple assignments small and this works well for our use case as we show in the experiments. The alternative of skipping the quadratic program and directly using a softmax over the parameters frequently results in degenerate assignments of large number of $P_1$ partitions assigned to a small number of $P_2$ partitions.

$$
\begin{aligned}
\text{minimize} \quad & \sum c_{uv} d_{uv} + \gamma \sum_{u,v} d_{uv}^2 + \gamma \sum_v p_v^2 + s_v^2 \\
\text{subject to} \quad & p_u + d_{u,v} + s_u = 1, \\
& p_v + d_{u,v} + s_v = 1 \ \{u, v\} \in E
\end{aligned}
\tag{7}
$$

## D  TRAINING DETAIL FOR CLEVR AND CLEVRTEX

We train using either a single A6000 GPU or 4 GTX 1080Ti GPUs for 1.5 days for ClevrTex and on Clevr using a batch size of 64. We use Adam with a learning rate of 4e-4. Training is generally stable

and we did not require learning rate warm-up. Although not strictly necessary, we trained with an exponential learning rate decay. For these experiments we used 12 slots with each slot having a size of 64.

# E  ABLATION EXPERIMENTS

## E.1  EFFECT OF INTERPIXEL EDGE WEIGHT

We perform an experiment to judge the effect of the interpixel edge weights on performance. We fix the interpixel edges weights to 0 and learn only the $s$ and $t$ edge weights. For this experiment we use the smaller *varied background* variant of ClevrTex downscaled to 64x64. We train the models with and without learning the interpixel edge weights for about 170 epochs with 10 slots. The results are shown in Table 3 with image masks shown in Figure 7. Without learning interpixel we see a large drop in segmentation performance of about 8 points in mIoU. The segmentation quality without learned edge weights is significantly poorer as also seen visually in Figure 7.

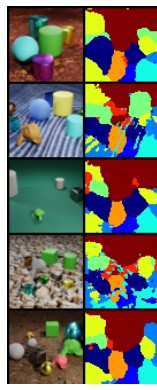 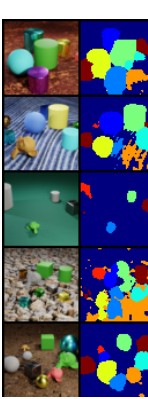

Table 3: Ablation results for learning interpixel weights

| Model | mIoU (%) | ARI-FG (%) |
|---|---|---|
| W/O Interpixel Weight | 19.4 | 52.7 |
| With Interpixel Weight | 27.3 | 67.1 |

Figure 7: Segmentation masks without learned interpixel weights (left) and with learned interpixel weights (right)

## E.2  EFFECT OF THE REGULARIZATION PARAMETER $\gamma$

Using the same setup as in the Section E.1, we perform experiments with various values for the regularization parameter $\gamma$. We choose $\gamma \in \{0.5, 1, 2\}$ and check evaluation set segmentation performance. Results are shown in Table 4. We find that quantitative performance on the evaluation set is best with $\gamma = 1$ and that performance significantly drops with large $\gamma$. Very small values of $\gamma$ can make training unstable since $\gamma$ appear inverted in 6 in the block term $C$.

Table 4: Effect of $\gamma$ on Segmentation Performance

| $\gamma$ | mIoU (%) | ARI-FG (%) |
|---|---|---|
| 0.5 | 27.3 | 67.1 |
| 1 | 30.6 | 68.7 |
| 2 | 15.2 | 49.7 |

