# OpenReview forum: "Differentiable Mathematical Programming for Object-Centric Representation Learning"
_ICLR.cc/2023/Conference — ICLR 2023 poster_

### Official Review · Reviewer_eQdL · 2022-10-19

**Confidence:** 5
**Correctness:** 3
**Technical Novelty And Significance:** 3
**Empirical Novelty And Significance:** 3
**Recommendation:** 8

**Clarity, Quality, Novelty And Reproducibility:**

The paper is well written and easy to understand. Every design choice is well explained. Therefore, it should be easy to reproduce the results, even without the code. Since the authors promise to provide the code, there are no reproducibility issues.

**Strength And Weaknesses:**

Strengths:
+ The method is well motivated and emphasizes the main challenges of the approach (differentiability, runtime and memory consumption)
+ The method builds nicely on existing methods like implicit function approach (Amos and Kolter), diff. mathematical programming (Wilder), graph cut in computer vision (Boykov, Kolmogorov), sparse Cholesky factorization (Chen et al.), etc.
+ The results are promising and demonstrate the usefulness of an LP layer in a deep learning framework

Weaknesses:
- The paper does not address the issue that the multi-cut problem is NP hard and that any solution can only be an approximation. Here, references to alpha-expansion or alpha-beta-swap [Boykov, Veksler, Zabih, ICCV'99] might be useful.
- Page 4 describes the graph not entirely correctly. Every vertex (except for source, sink and image boundary vertices) has 5 incoming and 5 outgoing edges, since edges between neighboring vertices are bi-directional.
- The experiments are a bit underwhelming, since semantic image segmentation experiments (CityScape) are missing

**Summary Of The Paper:**

The paper addresses the problem of turning the optimization of an LP (linear program) into a differentiable layer. This is done by approximating the LP by a QP (quadratic program) and using the QP as a differentiable proxy for the LP. The specific focus of this paper is the LP of finding an optimal s-t cut in a directed graph.

The s-t cut is then extended to a multi-cut problem which is used for object segmentation. Since the segmentation is done based on specific object features (used to define the graph), the approach does in fact perform an object-centric representation learning.

The paper explores how this representation learning helps in object discovery tasks and matching tasks.

**Summary Of The Review:**

Overall, the paper describes a nice approach on how to integrate s-t-cuts into a deep learning framework and demonstrates its usefulness for representation learning. The theoretical weaknesses are easy to fix for the final version. Due to the limiting experiments, I cannot give the maximal score. Nonetheless, I would like to see the paper presented at ICLR'23.

---

> ### Author Response · Authors · 2022-11-11
> **Author Reply**
>
> We thank the reviewer for the appreciative comments.
>
> >*Weaknesses:*
> >>    *The paper does not address the issue that the multi-cut problem is NP hard and that any solution can only be an approximation. Here, references to alpha-expansion or alpha-beta-swap [Boykov, Veksler, Zabih, ICCV'99] might be useful.*
>
> ## Multi-cut Clarification
> We would like to clarify that we do not intend to, or claim to, solve either the multiway cut or the $k$-cut or any other extension of the $s$-$t$ cut problem.
> We do solve $k$ parallel $s$-$t$ cut problems, which may bear a resemblance to $s$-$t$ cut generalizations, such as multicut.
> However, we emphasize that we only ever approximate the $s$-$t$ cut problem and combine parallel solutions with a softmax for the partition into $k$ components.
>
> It is also possible that confusion may arise from our use of the term $k$-partition.
> We used the term $k$-partition only to indicate a partition into $k$ components without any special restrictions on the components.
> Since the term $k$-partition also refers to special graph cut problems and can be misleading, we have replaced it in the updated draft.
>
> For further extension, it would indeed be interesting to consider generlizations of the problem and thank the reviewer for the references.
>
> >>    *Page 4 describes the graph not entirely correctly. Every vertex (except for source, sink and image boundary vertices) has 5 incoming and 5 outgoing edges, since edges between neighboring vertices are bi-directional.*
>
> ## Corrections
>
> In the referred paragraph we intended to say that per (x,y) coordinate we parameterize 6 edges (4 outgoing to neighbors, and s and t edges) with a convolutional feature map with 6 channels, rather than to describe the graph.
> Taken together these cover all edges.
> The formulation of the text was misleading and we have clarified it in the updated draft in Section 3.1.
>
>  >>   The experiments are a bit underwhelming, since semantic image segmentation experiments (CityScape) are missing
>
> ## Experiments
> Using graph cuts for segmentation can be interesting, however there is the question of whether it is essential or useful to differentiate through optimization solutions for this.
> The natural place for a cut operation in the context of semantic segmentation would be in the output layer.
> However, at the output a strong signal is provided by the labels.
> It is not immediately clear what advantage differentiability through the quadratic program solutions would afford in that case since one may be able to directly optimize the parameters.
>
> We focus on object-centric representations since here one can make a clear case for the usefulness of a differentiable partitioning operation deep inside the network's hidden layers.
>
> --
>
> We thank the reviewer for the review and the appreciation. Please let us know if there are any further questions or concerns.

---

> > ### Comment · Reviewer_eQdL · 2022-11-23
> > **Comment on Rebuttal**
> >
> > I like to thank the authors for their clarifications.
> >
> > Overall, I stay with my accept vote.

---

> > > ### Author Response · Authors · 2022-11-23
> > > **Author Response**
> > >
> > > We thank the reviewer for taking the time to read our response and for the recommendation.

---

### Official Review · Reviewer_wmJ4 · 2022-10-25

**Confidence:** 3
**Correctness:** 3
**Technical Novelty And Significance:** 2
**Empirical Novelty And Significance:** 2
**Recommendation:** 5

**Clarity, Quality, Novelty And Reproducibility:**

Overall clarity and quality is fair. The approach seems novel.

The notation is somewhat confusing. The overall loss function is not defined (Is it the quadratic program objective?). In (2), $s$ is used as both the slack variables and sink node. Unclear the details of solving KKT equations in Eq.6 (why solution is Ez and Cz?).

**Strength And Weaknesses:**

The interesting and promising aspect of the paper is that the initial minimum cut problem encodes pair-wise (topological) relationships between pixels.

Their actual formulation (2) is much different from minimum cut formulation in (1). By adding square regularization term and removing non-negativity constraints, the papers moves far from the original min cut. Their heuristic integration of k parallel s-t cuts is also quite far from classical definition of k-way cut. Though it still may be valuable, one can hardly use the seemed similarity to min cut for insights or explanations. It is a stretch to claim proposing graph cuts for k-partition with neural networks.

The paper lack ablation study of the pairwise weights. I.e. the experiment where all pairwise weights are set to 0 and only weights of edges connecting to s or t are learned.

The paper lack ablation study on gamma. How does performance of the method changes when changing gamma closer to 0?

**Summary Of The Paper:**

The paper propose a new aggregation method for Object-Centric Representation Learning. That is, a conversion of CxHxW feature map into KxHxW masks with K objects. Instead of standard attention of softmax, the paper injects an optimization problem related to minimum cut problem.

**Summary Of The Review:**

While overall approach seems novel and interesting, the significance of the results is not clear, see weakness section of the review.

---

> ### Author Response · Authors · 2022-11-11
> **Author Reply 2/2**
>
>
> >*The paper lack ablation study of the pairwise weights. I.e. the experiment where all pairwise weights are set to 0 and only weights of edges connecting to s or t are learned.*
> >
> >*The paper lack ablation study on gamma. How does performance of the method changes when changing gamma closer to 0?*
> >
>
> ## Ablations
> We have added ablation experiments for pairwise weights and for the effect of change in $\gamma$ in Appendix E in the updated draft.
> The experiments have been performed on the smaller 'varied background' variant of ClevrTex on images downscaled to 64x64.
>
> **Zero Pairwise Weight**.
> The results for the pairwise weights experiment are summarized in the following table.
>
> |Method|mIoU|ARI-FG|
> |----|----|----|
> |W/O Interpixel Weights|19.4|52.7|
> |With Interpixel Weights|27.3|67.1 |
>
> The result is that evaluation set segmentation performance drops significantly (about 8 points in mIoU) when the interpixel weights are not learned (set to 0) and only the $s$ and $t$ weights are learned.
> Visualizations of the learned masks can be seen in the updated paper which show a significant worsening of segmentation.
>
> **Changing $\gamma$**. The results for changes in the parameter $\gamma$ are shown in the following table.
>
> |$\gamma$|mIoU|ARI-FG|
> |----|----|----|
> |0.5|27.3| 67.1|
> |1|30.6|68.7 |
> |2|15.2| 49.7|
>
> We see that performance worsens significantly when $\gamma$ is set too large and an intermediate value gives the best results.
> Also we find in experiments that very small values of $\gamma$ can make training unstable.
> This can be explained by the presence of the factor of $1/\gamma$ in the term $C$ in equation 6 which becomes large when $\gamma$ is small.
>
> >>*The notation is somewhat confusing. The overall loss function is not defined (Is it the quadratic program objective?). In (2), is used as both the slack variables and sink node. Unclear the details of solving KKT equations in Eq.6 (why solution is Ez and Cz?).*
>
> ## Clarifications
>
> **Loss Function**. We have no new loss function for the models in the experiments.
> For the experiments we use the Slot Attention model in which we replace the slot attention module by our module.
> The loss function and the rest of the model is unchanged.
> In particular, the quadratic program does not appear in the objective, and is only solved inside the hidden layer it corresponds to.
> We have added a clarification in Section 4.
>
> **KKT equations**.
> The variable $z$ in $Cz$, $Ez$ in the paragraph after equation 6 is a dummy variable.
> We have renamed it to $z'$ to avoid confusion.
> During the forward pass $z'=c$ and we need to compute $Cc$ and $Ec$.
> During the backward pass $z'=g$, where $g$ is the incoming gradient and we need to compute $Cg$ and $Eg$.
>
> In further detail, solving the quadratic program requires solving the set of linear equations (equation 5) for $z$.
> This in turn requires inverting the matrix in equation 5.
> We rewrite the inverse in equation 6 in terms of factors $C$, $E$ etc by Gaussian elimination and solve by plugging into equation 5.
> This requires computing the factors $Cz'$, $Ez'$ (for some $z'$) which in turn require computing a large matrix inverse $(AA^t)^{-1}$, where $A$ is the constraint matrix.
> We precompute the Cholesky factors of $AA^t$ which allow us to solve the linear system in equation 5 without computing inverses during training.
>
> To obtain gradients, we solve the same system of equations in (5) except that $c$ is replaced by the incoming gradient $g$ and $b$ is replaced by $0$.
>
> Further details of computing solutions of equality constrained quadratic programs can be found in Nocedal and Wright (1999) and details of computing gradients can be seen in Amos and Kolter (2017) and Barratt (2019).
>
> **Notation**. The $s$ that corresponds to a source node in equation (2) is only an index rather a variable.
> However, to avoid confusion we have renamed the slack variable to $r$ in Section 3.1.
>
> --
>
> We hope to have addressed the reviewer's concerns. If there are other issues which require clarification, please let us know.

---

> ### Author Response · Authors · 2022-11-11
> **Author Reply 1/2**
>
> We thank the reviewer for the review and suggestions.
>
>
> >*Their actual formulation (2) is much different from minimum cut formulation in (1). By adding square regularization term and removing non-negativity constraints, the papers moves far from the original min cut. Their heuristic integration of k parallel s-t cuts is also quite far from classical definition of k-way cut. Though it still may be valuable, one can hardly use the seemed similarity to min cut for insights or explanations. It is a stretch to claim proposing graph cuts for k-partition with neural networks.*
>
> ## $k$-Partition Clarification
> We would like to clarify that we used the term $k$-partition only to indicate a partition into $k$ components without any special restrictions on the components.
> In particular we do not intend to, or claim to, approximate either the $k$-cut or multiway cut or any other extension of the $s$-$t$ cut problem.
> The $s$-$t$ cut problem is the only problem that we ever approximate and combine parallel solutions with a softmax for the partition into $k$ components.
> The use of the term $k$-partition may cause confusion so we have replaced it in the updated draft with the term $k$-part partition.
>
> ## Min-cut Approximation
> As for the formulation (2) being far from the minimum cut formulation, there are two changes in the formulation (2).
>
> 1. The addition of regularization terms in the objective
> 2. Removal of non-negativity constraints
>
> **Adding regularization terms**.
> If we only make the first change without removing non-negativity constraints, we  obtain a quadratic programming relaxation of the linear program (1) (See Wilder et al. (2018).
> This is a relaxation in the sense that the relaxed solution is always within an additive term, depending on $\gamma$, of the optimal as shown in Wilder et al. (2018).
> It is necessary to add the quadratic term because linear programs otherwise are not continuously differentiable in terms of the objective parameters.
>
> **Removing non-negativity constraints**.
> Our further approximation of removing the non-negativity constraints and replacing them by regularization also has precedent.
> The Quadratic Penalty Method (see Nocedal and Wright, Chapter 17) converts constrained quadratic programs into unconstrained quadratic programs.
> For each constraint the Quadratic Penalty Method adds a squared regularization term to the objective and repeatedly increases the regularization (penalty) coefficient.
> The main difference between the Quadratic Penalty Method and our method of regularized equality constrained programs is that we  add only the non-negativity constraints as regularization terms (instead of all constraints) and solve the resulting equality constrained program (instead of an unconstrained one).

---

### Official Review · Reviewer_miwq · 2022-10-28

**Confidence:** 1
**Correctness:** 4
**Technical Novelty And Significance:** 4
**Empirical Novelty And Significance:** Not applicable
**Recommendation:** 8

**Clarity, Quality, Novelty And Reproducibility:**

Overall, the contribution is clear, however, the structure and the presentation might be overworked:

The arrangement of figures, algorithms, and tables is hampering the reading flow.

The font size in Table 1 is too small. The table is hard to read.

The related work sections does not really discuss related work. This is just general an overview of works in this topic.

The mathematical writing must be seriously checked!

The bibliography entries needs to be seriously checked!

**Strength And Weaknesses:**

Strength:

The idea of approximating the original linear problem by a special quadratic program, which is differentiable is interesting.

In this way, graph cuts and modern neural networks can be combined in a reasonable way.


Weaknesses:

The experimental results are ambiguous. It is not fully clear for which kind of problems the approach would be meaningful in practice?

The structure and the presentation must be improved to increase the readability.

**Summary Of The Paper:**

The paper on hand presents an approach for s-t graph cuts, relying on a differentiable quadratic approximation of the original problem. The approach is demonstrated for different applications.

**Summary Of The Review:**

Even though the experimental results do not really show clear benefits, and also special cases are not discussed, the paper presents a fresh idea, going beyond just changing minor things in established methods. In particular, transferring a linear problem to a quadratic one, allowing to combine ideas from graph cuts with neural network learning is an interesting approach and a worthwhile contribution.

---

> ### Author Response · Authors · 2022-11-11
> **Author Reply**
>
> We thank the reviewer for the appreciation and suggestions.
>
> >>*The experimental results are ambiguous. It is not fully clear for which kind of problems the approach would be meaningful in practice?*
> >>
> >>*The structure and the presentation must be improved to increase the readability.*
> >>
> >>*Overall, the contribution is clear, however, the structure and the presentation might be overworked:*
> >>
> >>*The arrangement of figures, algorithms, and tables is hampering the reading flow.*
> >>
> >>*The font size in Table 1 is too small. The table is hard to read.*
> >>
> >>*The related work sections does not really discuss related work. This is just general an overview of works in this topic.*
> >>
> >>*The mathematical writing must be seriously checked!*
> >>
> >>*The bibliography entries needs to be seriously checked!*
>
>
> The approach in general is useful for such applications where it is useful to impose structural assumptions in hidden layers of the network.
> The focus of our work is on representation learning and on object-centric learning in particular which can benefit from a partitioning structure in hidden layers.
> Differentiability is essential for any method that imposes such strucuture in hidden layers and our method provides a scalable way to achieve this.
> The experimental results (Table 1) show that no other method works well across the textured and out-of-distribution input settings that we consider.
> Whereas our method is either the best performing or the next to best in almost all of the metrics that we consider.
>
> We have fixed the font size problem in Table 1 and have improved the bibliographic entries in the updated draft. We are also considering other structural changes for improved presentation for the final version.

---

### Official Review · Reviewer_9nSj · 2022-10-28

**Confidence:** 3
**Correctness:** 3
**Technical Novelty And Significance:** 2
**Empirical Novelty And Significance:** 2
**Recommendation:** 5

**Clarity, Quality, Novelty And Reproducibility:**

The submission provide enough details for reproducibility. for clarity and novelty, please see comments above.

**Strength And Weaknesses:**

Strengths:

- idea of Neural Graph cut is interesting. Experiments on two datasets show the effectiveness of the method wrt some baselines considered in the paper.

Weakness:

- Presentation: why do we need to study object centric representation learning. the submission should motivate it somewhere with few lines and contextualize it in broader context IMHO..e.g. how it relates to scene graph parsing approach  or constrained CNN (pathak et al. , ICCV 2015) which also optimizes a quadratic program with CNN to achieve semantic segmentation.

- Novelty: is there any technical novelty in section 3.1? maybe i am missing something but to me this section appears to be classic QP optimization..although the idea of  neural graph cut problem formulation is interesting, solution appears to be a straightforward extension of QP optimization..this is not a strong weakness but authors should clarify the novelty if any in this section. if not, much of this can be covered in a background section.

- Baselines: I did not see any baselines involving classical optimization schemes such as graph cut or normalized cut in the experiment section.  why not build an affinity graph with state of the art pretrained-CNN features on the two datasets and simply run a classical method. is scalability an issue here? besides, I did not find a scalability comparison with some existing approaches as one of the main claim of submission is its scalability to other approaches.

**Summary Of The Paper:**

This paper studies object centric representation learning and formulates it as a neural graph cut problem that involves parameterizing the coefficients of objective function of the linear program with neural networks. To this end, it converts the LP into a quadratic program that is easier to optimize with deep networks. The method is experimentally validated on object discovery as well as small scale matching tasks.

**Summary Of The Review:**

I am not sure about the significance of the problem considered in this paper and technical novelty of the solution proposed. i am happy to reconsider my rating based on rebuttal and other reviews.

---

> ### Author Response · Authors · 2022-11-11
> **Author Reply 2/2**
>
>
> >>
> >>*Baselines: I did not see any baselines involving classical optimization schemes such as graph cut or normalized cut in the experiment section. why not build an affinity graph with state of the art pretrained-CNN features on the two datasets and simply run a classical method. is scalability an issue here? besides, I did not find a scalability comparison with some existing approaches as one of the main claim of submission is its scalability to other approaches.*
>
> ## Classical Methods
>
> In this paper, we approach the problem of representation learning and, in particular, object-centric representation learning.
> Object-centric representation learning aims to learn hidden and internal representations of objects in a scene in neural network models and for this differentiability is essential.
> The method we propose in the paper is differentiable and scales to object-centric learning problems.
> On the other hand, classical methods, such as those for computing cuts over graphs, cannot be directly used for learning object-centric representations because they are not differentiable.
>
> Of course, once we have object-centric representations we also can use them for other tasks such as image segmentation.
> However, image segmentation in itself, is not the problem that we aim to solve in this paper and focus on learning object-centric representations.
>
> ## Scalability
> We would like to make a clarification about our claim of scalability.
>
> There has been prior work on differentiable optimization layers within neural networks.
> However, these methods are limited in the size of the problems that are solvable.
> For instance, the OptNet method (Amos and Kolter) is limited to problems with only about 1000 variables (see Section 3.3 in Amos and Kolter).
> To solve the $s$-$t$ cut problem for 64x64 size feature maps, however, we need to solve quadratic programs with over 50,000 variables and over 20,000 constraints and prior methods for differentiable optimization do not scale to such problems.
> Our claim of scalability is with reference to the use of fully differentiable application of mathematical programming approaches for image-scale data where previous approaches do not scale to such data while ours does scale.
>
>
> >>*Presentation: why do we need to study object centric representation learning. the submission should motivate it somewhere with few lines and contextualize it in broader context IMHO..e.g. how it relates to scene graph parsing approach or constrained CNN (pathak et al. , ICCV 2015) which also optimizes a quadratic program with CNN to achieve semantic segmentation.*
>
> ## Presentation
> We have added some motivation for object-centric representation learning in the introduction section in the updated draft.
> The main idea behind object-centric learning is to consider a complex scene as a composition of interacting but relatively independent entities.
> Object-centric representation learning aims to discover and represent this compostional structure.
> The potential goal is that the obtained representations generalize to a range of downstream tasks and improve robustness, sample efficiency and generalization (Elsayed et al. (2022)).
>
> The main difference between our method and methods employed by most other models using constrained optmization methods in computer vision is that our method works in a hidden layer of a deep model.
> The requires us to be able to  differeniate through the solution of the optmization problem.
> In contrast, a number of computer vision models including the reference given by the reviewer have constrained optimization problems in the output layer, where the gradients wrt the problem's parameter can be obtained from given labels, without the need to different through solutions of optimization problems.
>
> --
>
> We thank the reviewer for the review and hope that concerns have been addressed.

---

> ### Author Response · Authors · 2022-11-11
> **Author Reply 1/2**
>
> We thank the reviewer for reviewing our work.
>
> >>*Novelty: is there any technical novelty in section 3.1? maybe i am missing something but to me this section appears to be classic QP optimization..although the idea of  neural graph cut problem formulation is interesting, solution appears to be a straightforward extension of QP optimization..this is not a strong weakness but authors should clarify the novelty if any in this section. if not, much of this can be covered in a background section.*
>
> ## Novelty in 3.1
> The novelty in Section 3.1 is the approximation of the $s$-$t$ cut problem by a quadratic program with equality constraints only which is 1) differentiable relative to the objective function parameters and 2) efficiently solvable in batch for image data.
> To explain the importance of these novelties, we break them down as follows:
>
> - The exact formulation of the $s$-$t$ cut problem is in terms of a linear program.
> However, linear programs are not continuously differentiable relative to objective function parameters.
> The reason is that with linear programs the solution is always on one of the corners of the simplex, and small changes in the objective coefficients can lead to a sudden jump to a different simplex corner.
> Relaxing a linear program to a convex quadratic program (Wilder et al. (2018)) makes it differentiable, but the presence of inequality constraints (non-negativity constraints) makes solving them inefficient.
> - Methods for differential optimization in the literature do not scale to more than  a few hundred variables [Amos and Kolter (2017)].
> These methods are not being used directly on the image feature maps as we propose so with object-centric representations  but rather on smaller scale problems, whereas we need to solve programs with over 50,000 variables and 20,000 constraints in batch.
> To make the solution of the problem more efficient for parallel batch processing and application to raw pixel data, we remove non-negativity constraints (replacing them with regularization terms) so that we get quadratic programs with equality constraints only, which can be solved more efficiently in parallel than general quadratic programs.
> However, these programs still require large matrix inverses to be computed for both the forward and backward passes.
> - For the $s$-$t$ cut problem, we then note that we can avoid computing inverses with appropriate precomputation since the constraints do not change. For this we precompute the KKT matrix factorization with a sparse Cholesky factorization.
> Due to the precomputation, we do not need to compute inverses  during training since we can now solve the quadratic programs by sparse triangular solves.
> Interestingly, the same factorization can be used in the backward pass as well.
> - These optimizations taken together now allow us to scale to image data and to solve the corresponding quadratic programs that have with tens of thousands of variables and constraints.

---

### Author Response · Authors · 2022-11-11
**Summary**

We thank all reviewers for taking the time to review our work.
We have an updated draft of the paper based on the reviews with the following changes indicated in red in the updated draft.

1. We add an ablation experiment along with slot visualizations in Appendix E.1 to test the case where only the $s$ and $t$ edge weights are learned and inter-pixel edge weights are set to 0.
2. We add another ablation experiment in Appendix E.2 to test the effect of the regularization parameter $\gamma$ on performance.
3. General fixes and clarifications in Section 3.1 and Section 4 regarding notation and method description.
4. Improved the formatting of some bibliographic entries.

--

If there are any further questions we would be happy to address them.

---

### Author Response · Authors · 2022-11-18
**Further Discussion**

Dear Reviewers,

We are entering the last day of the discussion period. About six days ago we responded to each reviewer's points and uploaded a revised version of our paper which contains improvements motivated by the reviewers' points.

We would be grateful if the reviewers could take a look at our responses and update their review if we have cleared up the remaining concerns. If not, we are happy to incorporate further changes in the final version.

---

### Decision · Program_Chairs · 2023-01-20

**Decision:**

Accept: poster

**Justification For Why Not Higher Score:**

 The paper is quite strong, but still likely in a somewhat narrow niche area for the entirety of ICLR.  There were also a number of additional ablation experiments that, as the reviewers pointed out, could strengthen the work as it currently stands.

**Justification For Why Not Lower Score:**

 The paper presents a valuable contribution both to object-centric learning and a compelling application of differentiable optimization methods.  Overall these combine to make a compelling final contribution, and one definitely worth accepting.

**Metareview: Summary, Strengths And Weaknesses:**

Thank you for your submission to ICLR.  Although there was some disagreement amongst the reviewers, having gone through the paper and reviews myself, I agree with the more positive assessments of the paper.  In particular, I believe that the paper presents a notable contribution to the field, first in terms of a reasonable advance to the field in terms of representation learning (where the results are compelling in and of themselves).  But the work also represents a nice illustration of differentiable optimization and implicit layers themselves (with specialized algorithms for solving the optimization problems here), showcasing a valuable illustration of the power and potential of these approaches.  Overall, these two contribution together make a strong contribution.

Several reviewers did have some confusion regarding terminology and/or requested additional ablation experiments.  It would be great to address these clarification points in the camera ready version, plus adding the additional ablation experiments if possible.

**Note From Pc:**

if the above contains the word "oral" or "spotlight" please see: "oral" presentation means -> notable-top-5% and "spotlight" means -> notable-top-25%. As stated in our emails, we are disassociating presentation type from AC recommendations

**Summary Of Ac-Reviewer Meeting:**

 N/A